# NOTCH1-Related Leukoencephalopathy: A Novel Variant and Literature Review

**DOI:** 10.3390/ijms25052864

**Published:** 2024-03-01

**Authors:** Stefania Della Vecchia, Alessandra Tessa, Rosa Pasquariello, Luis Seabra, Yanick J. Crow, Roberta Battini

**Affiliations:** 1Department of Molecular Medicine and Neurogenetics, IRCCS Fondazione Stella Maris, 56128 Pisa, Italy; alessandra.tessa@fsm.unipi.it; 2Department of Neurosciences, Psychology, Drug Research and Child Health (NEUROFARBA), University of Florence, Viale Pieraccini, 6, 50139 Florence, Italy; 3Department of Developmental Neuroscience, IRCCS Fondazione Stella Maris, 56128 Pisa, Italy; rosa.pasquariello@fsm.unipi.it; 4Laboratory of Neurogenetics and Neuroinflammation, Institut Imagine, University of Paris, 75015 Paris, France; luis.seabra@institutimagine.org (L.S.); yanickcrow@mac.com (Y.J.C.); 5MRC Human Genetics Unit, Institute of Genetics and Cancer, University of Edinburgh, Edinburgh EH4 2XU, UK; 6Department of Clinical and Experimental Medicine, University of Pisa, 56126 Pisa, Italy

**Keywords:** heterozygous *NOTCH1* mutation, leukoencephalopathy with calcifications and cysts

## Abstract

*NOTCH1*-related leukoencephalopathy is a new diagnostic entity linked to heterozygous gain-of-function variants in *NOTCH1* that neuroradiologically show some overlap with the inflammatory microangiopathy Aicardi-Goutières syndrome (AGS). To report a 16-year-old boy harbouring a novel *NOTCH1* mutation who presented neuroradiological features suggestive of enhanced type I interferon signalling. We describe five years of follow-up and review the current literature on *NOTCH1*-related leukoencephalopathy. Clinical evaluation, standardised scales (SPRS, SARA, CBCL, CDI-2:P, WISCH-IV and VABS-2) and neuroradiological studies were performed, as well as blood DNA analysis. For the literature review, a search was performed on Pubmed, Scopus and Web of Science up to December 2023 using the following text word search strategy: (*NOTCH1*) AND (leukoencephalopathy). Our patient presents clinical features consistent with other reported cases with NOTCH1 mutations but is among the minority of patients with an onset after infancy. During the five-year follow-up, we observed an increase in the severity of spasticity and ataxia. However, at the age of 16 years, our proband is still ambulatory. As for other reported patients, he manifests psychiatric features ranging from hyperactivity during childhood to anxiety and depression during adolescence. The neuroradiological picture remained essentially stable over five years. In addition to the typical findings of leukoencephalopathy with cysts and calcifications already described, we report the presence of T2-hyperintensity and T1-hypotensity of the transverse pontine fibres, enhancement in the periventricular white matter after gadolinium administration and decreased NAA and Cho peaks in the periventricular white matter on MRS. We identified a novel heterozygous variant in *NOTCH1* (c.4788_4799dup), a frame insertion located in extracellular negative regulatory region (NRR)-domain as in previously published cases. Blood interferon signalling was not elevated compared to controls. This case provides further data on a new diagnostic entity, i.e., NOTCH1-related leukoencephalopathy. By describing a standardised five-year follow-up in one case and reviewing the other patients described to date, we outline recommendations relating to monitoring in this illness, emphasising the importance of psychiatric and gastroenterological surveillance alongside neurological and neuropsychological management. Studies are needed to better understand the factors influencing disease onset and severity, which are heterogeneous.

## 1. Introduction

*NOTCH1* is involved in essential cell signalling pathways with roles in inflammation and tumorigenesis. Germline heterozygous loss-of-function variants in *NOTCH1* are associated with Adams–Oliver syndrome (OMIM 90198) [1] and aortic valve disease 1 [2] (OMIM 109730), while somatic variants are implicated in carcinogenesis [3]. Recently, Helman and colleagues [4] described *NOCHT1* heterozygous gain-of-function variants to cause a new entity that neuroradiologically resembles the inflammatory microangiopathy seen in type I interferonopathies such as Aicardi–Goutières syndrome (AGS). In a series of seven cases, they described shared features of leukoencephalopathy, microangiopathy and calcifications, involvement of both the central and peripheral nervous system and elevated levels of the CXCL10 chemokine (also called IP-10) in the cerebrospinal fluid (CSF). In contrast to the majority of cases of AGS, evidence of enhanced interferon signalling in blood was not observed [4].

Here, we report a male with a novel *NOTCH1* (NM_017617.5) heterozygous variant (c.4788_4799dup; p.Ser1597_Leu1600dup) located in the extracellular negative regulatory region domain of the protein, delineating his neuroradiological, clinical and psychopathological profile over five years of follow-up. Based on a review of the literature, we summarise the clinical spectrum of *NOTCH1*-related leukoencephalopathy as currently described.

## 2. Methods

### 2.1. Case Report

Whole exome sequencing (WES) was performed on NovaSeq6000 system (Illumina, San Diego, CA, USA) using SureSelect Human All Exon V.7 kit (Agilent Technologies, Santa Clara, CA, USA). Generated reads were aligned to the human genome assembly hg19, and we used enGenome eVai software (evai.engenome.com, accessed on 9 December 2022) for the annotation and the interpretation of variants. We focused on rare variants with a read depth ≥30, MAF (minimum allele frequency) <1% in 1000 Genome Project, (www.1000genomes.org, accessed on 1 March 2023), ExAc (http://exac.broadinstitute.org, accessed on 1 March 2023) and gnomAD (https://gnomad.broadinstitute.org, accessed on 1 March 2023) databases predicted to be damaging for protein function (PolyPhen-2, SIFT, Mutation Taster, CADD-phred prediction tools). American College of Medical Genetics and Genomics criteria were used to classify variants. 

Sanger sequencing analysis was performed to confirm the presence of the identified variants in the proband and to perform segregation analyses in family members. 

We recorded clinical data and used standardised scales to define the neurological, neuropsychological and psychopathological course of the disease over time in our proband. For the neurological assessment, we used the Spastic Paraplegia Rating Scale (SPRS) [5], the Spatax Disability Scale [6] and the Scale for the assessment and rating of ataxia (SARA) [7]. We used the Wechsler Intelligence Scale for Children (WISC)-IV) [8] and the Vineland Adaptive Behaviour Scales 2nd edition (VABS-II) [9] to assess cognitive profile and adaptive skills. For psychopathological profiling, we used the Child Behaviour Checklist 6–18 years [10] and the parent version of the CDI-2 [11]. We also describe a five-year follow-up of brain imaging, recorded using 1.5 or 3 Tesla scans, with the following sequences: three-dimensional (3D) T1-weighted (T1W), SWAN (susceptibility-weighted angiography), 3D T2 fluid-attenuated inversion recovery and two-dimensional (2D) T2-weighted (T2W) fast spin echo.

Informed consent was provided by the mother as the legal representative of the proband.

### 2.2. Literature Review

We conducted an electronic database search in Pubmed, Scopus and Web of Science, up to December 2023 using the following text word search strategy: (*NOTCH1*) AND (leukoencephalopathy). We screened all titles and abstracts (Pubmed *n* = 11, Scopus *n* = 17, Web of Science *n* = 9) in order to exclude clearly irrelevant articles. After the database search, the reference lists of the included articles were examined for possible articles that were not located in the database search. We included in the review all articles reporting on the clinical and/or neuroradiological phenotype of *NOTCH1* heterozygous gain-of-function variants, with no restrictions on the age or sex of the participants. However, we excluded articles that: (i) did not provide sufficient clinical information; (ii) were written in a language other than English or Italian; (iii) were conference abstracts or reviews and (iv) were clearly not related to our topic.

## 3. Results

### 3.1. Case Report

Our proband is a now 16-year-old male, the second child of nonconsanguineous parents. The family history is negative for neuropsychiatric disorders. He was born at term after an uneventful pregnancy. His birth weight was 3110 g, his length was 49 cm and his occipitofrontal circumference (OFC) was 36 cm. He acquired independent ambulation at 12 months of age, produced his first words around 12 months of age and spoke his first sentences at around 2 years of age. In preschool he was described as a hyperactive child. At around six years of age, he began to exhibit school difficulties and clumsiness, and after a short time, he was noted to walk with a stiff gait. When he arrived at our centre, at the age of 12.5 years, he demonstrated spastic-paraparesis with sensorimotor neuropathy, ataxia, bradykinesia, moderate intellectual disability and strabismus. A megacolon was diagnosed, but so far, we do not know whether it can be defined as Hirschsprung’s disease. We monitored his neurological, neuropsychological and psychiatric features over time. Standardised neurological scales, cognitive profile and psychiatric questionnaires are reported in Table 1. SPRS (range 0–52) showed a worsening of spasticity severity at 16 (14.5/52) compared to 12.5 years of age (11/52). The spatax disability scale (range 0–7) showed an almost stable functional picture, although the need for walking orthoses became manifest at 16 years of age. SARA scale was only assessed at 16 years of age, giving a score of 10/40. Administration of the WISC-IV scale at 12.5 years revealed moderate intellectual disability with a harmonious profile between verbal and performance skills, confirmed at age 16 years (see Table 1). VABS-II indicated moderately impaired adaptive abilities at both 12.5 years and 16 years of age in all areas investigated (communication, socialisation and daily living skills (Table 1)). Regarding psychiatric manifestations, the CBCL 6/18 questionnaire at 16 years of age showed clinical scores in features related to affective problems, anxiety, aggressive behaviours and hyperactivity (Table 1). CDI-2 confirmed the presence of depressive symptoms (Table 1), with valproic acid initiated because of the presence of these features.

The neuroradiological phenotype observed in our patient is characterised by leukoencephalopathy with calcifications and microcysts. Figure 1 shows aspects of brain MRI imaging over time. The five-year neuroradiological follow-up indicated no morphological change over this time. There was a little change in gadolinium enhancement at the age of 14 years compared to 13 years, with a reduction in enhancement especially in the white matter. Decreased N-acetylaspartate (NAA) and choline (Cho) peaks in the PV white matter was present on magnetic resonance spectroscopy (MRS).

Our first diagnostic hypothesis was an atypical form of AGS characterised clinically by complex spastic paraparesis and radiologically by brain calcifications, leukoencephalopathy and cerebral atrophy, as previously described [12,13]. However, the analysis of blood interferon signalling and AGS-related genes in our proband was negative. Thus, we analysed a panel of leukoencephalopathy-associated genes that allowed us to exclude other leukoencephalopathies with cysts and calcifications [14] without reaching a genetic diagnosis. We then performed an analysis of mitochondrial DNA and mitochondrial-associated genes without obtaining significant results. The re-examination of exome sequencing data following the publication of Helman et al. [4] allowed us to reach a diagnosis through the identification of a c.4788_4799dup *NOTCH1*-variant in our proband. This novel variant is a frameshift insertion located in extracellular negative regulatory region (NRR)-domain as in the series published by Helman et al. [4]. This variant is absent in gnomAD (https://gnomad.broadinstitute.org/, accessed on 1 March 2023) and other public databases and was not detected in the child’s healthy mother and brother. Paternal DNA was unavailable.

### 3.2. Literature Review and Clinical and Neuroradiological Spectrum of NOTCH1-Related Leukoencephalopathy

Table 2 and Appendix A, respectively, describe clinical and neuroradiological features of *NOTCH1* patients included in the review.

We found only two papers describing this new entity. Helman and colleagues [4] described a case-series of seven patients, followed by a case report by Nicita and colleagues [15]. To date, we know that disease onset is variable, with 66% of cases manifesting in infancy (during the first year of life) and 33% at a later age, ranging from 6 to 40 years. The median age at the last evaluation was 16 years (range 2–65 years). Cognitive decline was present in all patients described, followed by spasticity (89%), developmental delay (67%), extrapyramidal signs (44%) and psychiatric manifestations (44%). Other possible signs included hypotonia (33%), ataxia (33%), peripheral neuropathy (22%) and epilepsy (22%). Hirschsprung disease has been reported in 33% of cases. Concerning walking, 3/9 never achieved autonomous ambulation, while 4/9 required a wheelchair at a median age of 12.5 (range 6–25) years. Considering patients with psychiatric manifestations (4/9, 44%), mood disorders (particularly depression) were reported in all patients, followed by aggressive behaviour, anxiety and hyperactivity (Figure 2). Analysis of brain MRI was available for all patients and revealed altered structural neuroimaging in 100% of cases: leukoencephalopathy, brain calcifications and atrophy were present in all patients. In addition, in some cases brain cysts and thinning of the corpus callosum were reported (Figure 2). Figure 2 summarises the clinical and neuroradiological phenotype of patients with *NOTCH1* heterozygous gain-of-function variants.

## 4. Discussion

The first report of *NOTCH1*-related leukoencephalopathy described seven unrelated patients who developed progressive leukoencephalopathy with calcifications associated with clinical involvement of both the central and peripheral nervous systems [4]. This was followed, shortly afterwards, by a description of one Italian child who showed a similar clinical evolution and also presented white matter disease, brain cysts and calcifications [15]. To date, *NOTCH1* heterozygous variants have been reported in a total of nine patients, including our proband.

From a clinical point of view, the condition is characterised by a variable age of onset, ranging from infancy to adulthood, with the earliest cases manifesting in the first year of life and later ones in a range of 6 to 40 years. The infantile cases are characterised by failure to achieve some cognitive and motor milestones, hypotonia, motor signs and sometimes epilepsy. Later-onset cases, on the other hand, present with a spastic or spastic-ataxic gait, often with a peripheral neuropathy, cognitive impairment and psychiatric manifestations. Concerning walking, some children never achieve autonomous ambulation, and 44% of patients are confined to a wheelchair by the median age of 12.5 years (range 6–25 years). Our proband demonstrates typical clinical features of the disease and is among the minority of patients with an onset beyond infancy. During a five-year follow-up, we observed an increase in the severity of spasticity and ataxia. However, at the age of 16 years, our proband is still ambulatory.

Global developmental delay, intellectual disability or cognitive impairment are present in all patients and often accompanied by psychiatric manifestations. Among the psychiatric disorders, depression and aggressive behaviours are the most frequent and adversely affect adaptive abilities and family daily life. For this reason, it is important to recognise and treat the emotional manifestations appropriately, with the help of dedicated professionals such as home educators. Our patient showed an almost stable cognitive level over time (moderate intellectual disability) but presented psychiatric manifestations including hyperactivity at pre-school age to anxiety and depression during adolescence.

Brain imaging is reminiscent of enhanced type I interferon signalling, as seen in AGS. The neuroradiological phenotype seems to be slowly progressive and dominated by leukoencephalopathy, calcifications and brain atrophy. In some cases, cerebral cysts and the thinning of the corpus callosum have also been reported. In our proband, we observed a neuroradiological picture that was stable over a period of five years. In addition to the neuroimaging features described so far, we also observed the presence of T2-hyperintensity and T1-hypointensity of the pontine transverse fibers, enhancement in the periventricular white matter and other cerebral regions after gadolinium administration and decreased NAA and Cho peaks in the periventricular white matter on MRS. Calcifications appear to be at least partly related to chronic inflammation [4]. The decrease in NAA and Cho peaks on MRS observed in our case could indicate, as described in the leukoencephalopathy with calcifications and cysts (LCC) [16], that the leukoencephalopathy is due to increased myelin water content rather than demyelination. Notch signalling plays crucial roles in adult brain function, including synaptic plasticity, memory and olfaction. Several lines of evidence suggest an involvement of this pathway in neurodegeneration, possibly through functional interactions with receptors such as Reelin, apolipoprotein E receptor 2 (ApoER2), N-methyl-D-aspartate receptor (NMDAR), CREB [17] and Arc/ARg3.1 [18,19]. Furthermore, the interaction of NOTCH with Delta/Notch-like epidermal growth factor (EGF)-related receptor (DNER)—a transmembrane protein specifically expressed in dendrites and cell bodies of postmitotic neurons—with a role in neural progenitor development, neural proliferation, neuronal and glial differentiation [20,21]—has been implicated in neurodegeneration by influencing synaptic transmission [22].

The most frequent extra-neurological manifestation appears to be Hirschsprung disease. It is therefore important to consider a multidisciplinary surveillance, including gastroenterological and surgical evaluation, particularly in those cases with constipation and abdominal distention.

Since NOTCH1 signalling plays a role in T- and B-cell development, and its somatic variants are associated with T- and B-cell leukaemia, Helman and colleagues [4] analysed the phenotype and function of T- and B-cells in their case series and found an imbalance in the development of peripheral B-cells, while the phenotype and cellular activities of T-cells were unaffected. Unfortunately, we did not verify these features in our proband.

Due to possible overlap with AGS, we studied the interferon signature in the peripheral blood of our case and found normal values—as reported previously [4]. In one case, CXCL9 and CXCL10 were found to be increased in the blood compared to controls, although lower than those of AGS patients, so the authors suggested their measurement as an additional diagnostic tool for this condition [15]. In contrast, Helman et al. found an increase in CXCL10, a cytokine secreted by astrocytes in response to IFN subtypes downstream of NOTCH1 signalling, only in CSF and not in plasma [23]. Therefore, the plasma and CSF inflammatory profile of this novel disease remains to be defined.

In conclusion, leukoencephalopathy with cysts and calcifications related to heterozygous *NOCTH1* variants is a recently described chronic inflammatory leukoencephalopathy showing some neuro-radiological overlap with certain type I interferonopathies. We identified a novel *NOTCH1* variant that expands the molecular spectrum of the disease and provides further clinical information relating to this new diagnostic entity. Our patient demonstrates clinical features as already reported, although with so few cases described, the breadth of phenotype remains to be defined. At the age of 16 years, he is still ambulant, and his cognitive and adaptive skills have remained stable over the past five years. By describing a standardised five-year follow-up in our case, and by reviewing all other cases described to-date, this work outlines the main areas to be monitored in this illness, emphasising the importance of psychiatric and gastroenterological surveillance alongside neurological and neuropsychological monitoring. Studies are needed to better understand the factors influencing disease onset and severity, as these seem to be widely heterogeneous.

The major strength of our study is the summary of the main clinical and neuroradiological features of the recently described *NOTCH1*-related leukoencephalopathy, which can help clinicians diagnose this condition and monitor it over time. While we describe only a single case, our report provides extensive follow-up data. Obviously, we expect that over time new information will help to better define the progression of the disease, and how the disease course differs depending on the age of onset. The main limitations of this study include the limited possibility of generalising the results on the basis of a single case, the small number of patients published to date and the lack of a detailed analysis of blood cells and cytokines in the blood and CSF of our case.

## Figures and Tables

**Figure 1 ijms-25-02864-f001:**
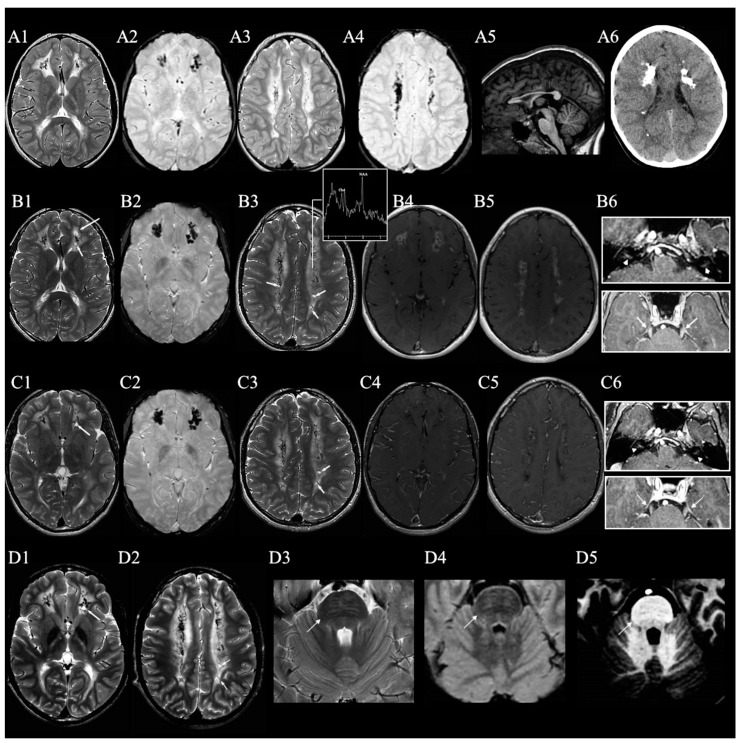
Five-year neuroradiological follow-up of the proband. (**A1**–**A6**) (1.5 T) were acquired at the age of 10 years, (**B1**–**B6**) (1,5 T) at the age of 13 years, (**C1**–**C6**) (1.5 T), at the age of 14 years and (**D1**–**D5**) (3 T) at the age of 16 years. The neuroradiological phenotype is characterised by leukoencephalopathy with calcifications and microcysts. The white matter lesions are hyperintense on T2 weighted imaging and localised in the periventricular (PV) white matter, corona radiata, centrum semiovale, posterior arm of the internal capsule and corpus callosum (CC). The CC is thinned, especially in the trunk (**A5**). Numerous calcifications tending to confluence, confirmed on CT scan (**A6**), and rare microcysts (see arrows in (**B1**,**B3**,**C1**,**C3**)) are present in the abnormal white matter. Neuroradiological follow-up (**C1**) four years after the first examination (**A1**) showed no further changes. After gadolinium administration, enhancement was observed in the PV white matter (**B4**,**B5**), in the intracisternal tract of the trigeminal nerve (see arrowheads in **B6 top**), and in the meatal tract of the acoustic nerve (see arrows in **B6 bottom**). At follow-up, a reduction in enhancement was observed, especially in the white matter (**C4**,**C5**). (**C6 top** and **C6 bottom**) show respectively the follow-up of gadolinium administration in the intracisternal tract of the trigeminal nerve (see arrowhead in **C6 top**), and in the meatal tract of the acoustic nerve (see arrows in C6 bottom). Decreased N-acetylaspartate (NAA) and choline (Cho) peaks in the PV white matter was present on magnetic resonance spectroscopy (MRS) (**B3**). The last brain MRI (**D1**–**D5**), performed at a higher magnetic field strength than the previous ones (3 T), shows neuroradiological findings overlapping with the previous imaging. The presence of T2-hyperintensity and T1-hypointensity of the pontine transverse fibres is evident, probably due to the greater resolution of the scans. Indeed, retrospectively, this finding was already identifiable in earlier sequences.

**Figure 2 ijms-25-02864-f002:**
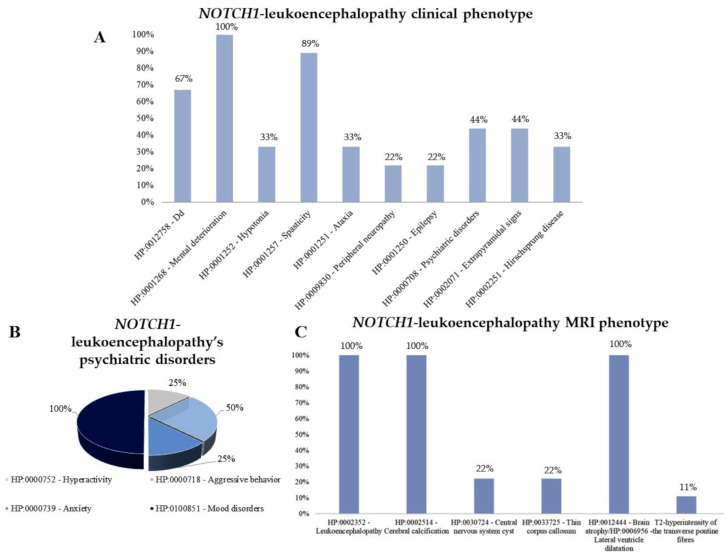
Clinical and neuroradiological phenotype of patients with *NOTCH1* heterozygous gain-of-function variants (*n* = 9). (**A**) Clinical spectrum of NOTCH1-leukoencephalopathy. (**B**) Psychiatric disorders found in patients with NOTCH1-leukoencephalopathy. (**C**) MRI phenotype observed in patients with NOTCH1-leukoencephalopathy.

**Table 1 ijms-25-02864-t001:** Standardised neurological scales, cognitive profile and psychiatric questionnaire of our proband. CBCL: Child Behaviour Checklist. CDI-2:P: Children’s Depression Inventory 2nd Edition, Parent.

	12.5 Years	16 Years
**Standardised neurological assessment**		
SPRS (0–52)	11	14.5
Spatax disability index (0–7)	3	3
SARA (0–40)	7	10
**Adaptive skills (VABS)**		
Communication (IQ deviation)	49	57
Socialisation (IQ deviation)	51	47
Daily living skills (IQ deviation)	44	55
**Cognitive assessment (WISC-IV)**		
ICV	54	50
IRP	50	43
IML	64	49
IVE	47	50
**CBCL 6–18 years**		
Syndrome scales	-	Anxious/Depressed, T = 85 Withdrawn/Depressed, T = 86 Somatic Complaints, T = 78 Social Problems, T = 77 Thought Problems, T = 69 Attention Problems, T = 86 Rule-Breaking Behaviour, T = 67 Aggressive Behaviour, T = 69
DSM-Oriented scales	-	Affective Problems, T = 89 Anxiety Problems, T = 77 Somatic Problems, T = 68 Attention Deficit /Hyperactivity Problems, T = 77 Oppositional Defiant Problems, T = 69 Conduct Problems, T = 66
**CDI-2:P**	-	Total Score, T = 71 Emotional Problems, T = 81 Functional Problems, T = 58

**Table 2 ijms-25-02864-t002:** Clinical features of *NOTCH1* patients and relative studies including in the review.

Ref	Onset	Current Age	DD	Gait Problems	Latest Examination	Psychiatric Features	Other	Variant (NM_017617.5)	ACMG/AMP
**This study**	6 yr	16 yr	**N**	6 yr	Spastic-paraparesis, bradykinesia, ataxia, sensorimotor neuropathy, intellectual disability	Hyperactivity in preschool, anxiety, depressed mood and repetitive behaviors	Strabismus and probable HD	c.4788_4799dup, p.Ser1597_Leu1600dup	VUS
**Nicita et al.** [15]	1 mo	2 yr	Y	NAIA	Poor cognitive and motor development, truncal hypotonia, appendicular spasticity, exaggerated startle response, fever-induced focal motor seizures from 9 months	None	Congenital macrocephaly, ureteropelvic junction stenosis	c.4811 T > G, p.(Val1604Gly)	LP
**Helman et al.** [4]	1 yr	25 *	Y	2 yr, wheelchair at 6 yr	Intellectual impairment, spasticity, ataxia and peripheral neuropathy	Cognitive decline, neuropsychiatric symptoms with passive disposition, lack of movement and communication and mood fluctuations	Hypogonadotropic hypogonadism	c.4814_4819dupTCTTCA p.(Phe1606_Lys1607insIlePhe)	LP
	15 yr	37 *	N	20 yr, wheelchair at 25 yr	Appendicular ataxia, severe spasticity, pseudobulbar dysarthria, peripheral neuropathy	Cognitive decline	Type 2 Diabetes	c.5046C > G p.(Asn1682Lys)	LP
	40 yr	65 yr	N	40 yr	Bilateral leg weakness, spasticity, imbalance	Depression and cognitive decline	Episode of myelitis treated with steroids in adolescence	c.4583G > A p.(Cys1528Tyr)	LP
	1 yr	28 yr	Y	1 yr, wheelchair at 16 yr	Lost speech, spasticity, dystonic foot posturing, painful muscle spasms	Depression, temper issues, periods of aggression, inappropriate laughing and crying	Staring spells but no epileptic EEG activity	c.5046_5047delCCInsAG p.(Asn1682_Arg1683insLysGly)	P
	0 yr	9 yr	Y	5 yr, wheelchair at 9 yr	Intellectual disability, spasticity, global muscle wasting	No	HD, short stature	c.5042A > G p.(Asp1681Gly)	P
	0 yr	6 yr	Y	NAIA	Intellectual disability, truncal hypotonia, limited head control, appendicular spasticity, dystonic movements	No	HD, atrial septal defect, glabellar hemangioma, microcephaly, short stature	c.5078T > C p.(Phe1693Ser)	P
	0 yr	6 yr	Y	NAIA	Intellectual disability, generalised hypotonia, myoclonic seizures	No	No	c.4754T > C p.(Leu1585Pro)	P

* Age at last examination before death; yr, years; Y, yes; N, no; NAIA, never achieved independent ambulation. DD: developmental delay. HD: Hirschsprung disease. VUS: Uncertain significance. P: Pathogenic. LP: Likely Pathogenic.

## Data Availability

Data Availability Statements are available upon request from the corresponding author.

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
