# Peer review of "NOTCH1-Related Leukoencephalopathy: A Novel Variant and Literature Review"

_ijms, 2024, doi:10.3390/ijms25052864_

Round 1

Reviewer 1 Report

Comments and Suggestions for Authors

The clinical case described in the manuscript is interesting. The case description may be helpful for other research and medical practice. However, the authors should better describe the clinical case. The description should include characteristics of blood morphology (what is known about CD4(+) and CD8(+) cells), selected metabolic parameters. What is known about the function of the receptor for neuronal DNER? What are the consequences of the presence of a genetic variant on the splicing forms of the NOTCH1 transcript and changes in the protein level (what changes in the 3- and 4-row structure of NOTCH1 does this genetic change cause).

Author Response

REVIEWER 1

The clinical case described in the manuscript is interesting. The case description may be helpful for other research and medical practice. However, the authors should better describe the clinical case.

  1. The description should include characteristics of blood morphology (what is known about CD4(+) and CD8(+) cells), selected metabolic parameters.

We thank the reviewer for this suggestion, noting the comment in the paper by Helman and colleagues (doi.org/10.1002/ana.26477) that: "The known impact of NOTCH signaling on T- and B-cell development, association of somatic NOTCH1 variants with T- and B-cell leukemia, and increased CSF IP-10 levels prompted testing T- and B-cell phenotype and function. We found an imbalance in peripheral B-cell development, but T-cell phenotype and cellular activities appeared unaffected. Unfortunately, for practical reasons, we do not have the possibility to collect further blood samples in the short-term, but will consider this when our patient is next seen for review.

  1. What is known about the function of the receptor for neuronal DNER?

We thank the reviewer for this question. We have now added a comment in the Discussion about the possible role of NOTCH1-DNER in neurodegeneration (see lines 253-258).

  1. What are the consequences of the presence of a genetic variant on the splicing forms of the NOTCH1 transcript and changes in the protein level (what changes in the 3- and 4-row structure of NOTCH1 does this genetic change cause).

Again, we thank the reviewer for this question. We have not undertaken any formal studies of splicing or structural modelling because the p.(Val1599_Leu1600insProAlaAlaCys) is an in-frame insertion, with no predicted impact on splicing according to in silico analyses (SpliceAI, SPiP, ESEfinder, RESCUE-ESE and EX-SKIP), and - given the overlap in phenotype with the other cases published to date - likely confers a gain-of-function.

Reviewer 2 Report

Comments and Suggestions for Authors

The authors reported an interesting description of a case report of NOTCH1-Related Leukoencephalopathy and a summary of the literature.  The paper is relevant in the field since expand the current knowledge of the genetic variability in NOTCH1 patients.  The comparison with other published materials in the literature is well done.  The paper is well written, conclusions are consistent with the evidence and arguments presented in the manuscripts.  Figures and Tables are very explicative for the readers.

I have only a comment regarding the Methods Sections:

- As far as the literature review is concerned, I would suggest to expand the keywords that the authors have used because 11 articles for a literature search is a very small number.  I understand the rare frequency of those disorders, but maybe a broader search may strength the value of the paper.

Author Response

REVIEWER 2

The authors reported an interesting description of a case report of NOTCH1-Related Leukoencephalopathy and a summary of the literature.  The paper is relevant in the field since expand the current knowledge of the genetic variability in NOTCH1 patients.  The comparison with other published materials in the literature is well done.  The paper is well written, conclusions are consistent with the evidence and arguments presented in the manuscripts.  Figures and Tables are very explicative for the readers.

I have only a comment regarding the Methods Sections:

  1. As far as the literature review is concerned, I would suggest to expand the keywords that the authors have used because 11 articles for a literature search is a very small number.  I understand the rare frequency of those disorders, but maybe a broader search may strength the value of the paper.

We thank the reviewer for this comment. The total number of search articles from the different databases (PubMed, Web of science and Scopus) is 38 articles. An expansion of the Keyword search (for example, using only NOTCH1) results in too broad a selection of papers – most of which are far removed from the our focus on NOTCH1 leukoencephalopathy.

Reviewer 3 Report

Comments and Suggestions for Authors

At the manuscript “NOTCH1-Related Leukoencephalopathy: A Novel Variant and Literature Review” by Drs. Stefania Della Vecchia et al authors reported a patient carrying a novel NOTCH1 mutation and describe 5 years of observations. The presented manuscript is devoted to leukoencephalopathy associated with the NOTCH1 peptide. The authors discovered a mutation in the NOTCH1 gene in a patient who demonstrated neuroradiological features suggestive of increased type I interferon signaling.

The value of the work lies in the long-term follow-up of the patient using several standard scales (SPRS, SARA, CBCL, CDI-2:P, WISCH-IV and VABS-2), as well as neuroradiological examination and DNA analysis. Such a comprehensive study certainly helps to bridge the gap between the analysis of clinical manifestations (i.e., phenotypic features) and the characteristics of the patient’s genotype. Such a gap is quite typical for rare hereditary diseases.

The authors also conducted a major review of the current literature on NOTCH1. Very interesting and rare data was shared, and I have no objections to this part of the manuscript. The review part is also carefully executed and informative, but I would like to clarify some questions.  These data certainly add much to what is already known regarding the phenotypic manifestations of NOTCH1 gene variations in clinical practice.

It is known that neuronal NOTCH signaling is regulated by another protein, called Arc/Arg3.1 - activity-induced gene. It was shown that in Arc/Arg3.1 mutants’ proteolytic activation of NOTCH1 is disrupted.

The authors did not mention the relationship on the part of the Arc protein, I think this needs some attention. I suggest to cite Sibarov et al, Front Neurol. 2023 7:14:1201104. doi: 10.3389/fneur.2023.1201104 and some other modern publications on this topic. This would seriously strengthen the manuscript.

I have no serious recommendations regarding improvements methodology, since long-term observation of a patient is in itself a labor-intensive process. Of course, there is a set of effective MRI methods that could provide a certain amount of additional data, but when obtaining data from a single patient, these methods cannot always be applied. And those methods that the authors were able to use, they used with maximum efficiency.

The presented conclusions are consistent with the studied material. Regarding the literature review and discussion, I wrote my advice above. The list of references is quite adequate, the figures and tables fully comply with generally accepted requirements. The presentation of a subject is systematic and comprehensive and analysis is proper. I am happy to recommend the manuscript for the publication after minor corrections mentioned above.

Author Response

REVIEWER 3

At the manuscript “NOTCH1-Related Leukoencephalopathy: A Novel Variant and Literature Review” by Drs. Stefania Della Vecchia et al authors reported a patient carrying a novel NOTCH1 mutation and describe 5 years of observations. The presented manuscript is devoted to leukoencephalopathy associated with the NOTCH1 peptide. The authors discovered a mutation in the NOTCH1 gene in a patient who demonstrated neuroradiological features suggestive of increased type I interferon signaling.

The value of the work lies in the long-term follow-up of the patient using several standard scales (SPRS, SARA, CBCL, CDI-2:P, WISCH-IV and VABS-2), as well as neuroradiological examination and DNA analysis. Such a comprehensive study certainly helps to bridge the gap between the analysis of clinical manifestations (i.e., phenotypic features) and the characteristics of the patient’s genotype. Such a gap is quite typical for rare hereditary diseases.

The authors also conducted a major review of the current literature on NOTCH1. Very interesting and rare data was shared, and I have no objections to this part of the manuscript. The review part is also carefully executed and informative, but I would like to clarify some questions.  These data certainly add much to what is already known regarding the phenotypic manifestations of NOTCH1 gene variations in clinical practice.

I have no serious recommendations regarding improvements methodology, since long-term observation of a patient is in itself a labor-intensive process. Of course, there is a set of effective MRI methods that could provide a certain amount of additional data, but when obtaining data from a single patient, these methods cannot always be applied. And those methods that the authors were able to use, they used with maximum efficiency.

The presented conclusions are consistent with the studied material. Regarding the literature review and discussion, I wrote my advice above. The list of references is quite adequate, the figures and tables fully comply with generally accepted requirements. The presentation of a subject is systematic and comprehensive and analysis is proper. I am happy to recommend the manuscript for the publication after minor corrections mentioned above.

  1. It is known that neuronal NOTCH signaling is regulated by another protein, called Arc/Arg3.1 - activity-induced gene. It was shown that in Arc/Arg3.1 mutants’ proteolytic activation of NOTCH1 is disrupted. The authors did not mention the relationship on the part of the Arc protein, I think this needs some attention. I suggest to cite Sibarov et al, Front Neurol. 2023 7:14:1201104. doi: 10.3389/fneur.2023.1201104 and some other modern publications on this topic. This would seriously strengthen the manuscript.

We thank the reviewer for this comment. We have now added comment in the Discussion (see lines 248-253) relating to the possible involvement of NOCTH1 signalling in dementia.

Round 2

Reviewer 1 Report

Comments and Suggestions for Authors

The authors responded to my comments but did not address most of them in their manuscript. The authors should clearly describe all the limitations of their study and interpretation of the results.

Author Response

We thank the reviewer for this question.

We have added a comment in the Discussion about the role of NOTCH1 signaling on T- and B-cell development (see lines 263-267).

We didn’t add in the manuscript the impact of our variant on splicing because the variant is an in-frame insertion, with no predicted impact on splicing.

Then we have added the main strengths and limitations of our study (see lines 291-300).